# The Group A Streptococcal Vaccine Candidate VAX-A1 Protects against Group B *Streptococcus* Infection via Cross-Reactive IgG Targeting Virulence Factor C5a Peptidase

**DOI:** 10.3390/vaccines11121811

**Published:** 2023-12-03

**Authors:** Sinead McCabe, Elisabet Bjånes, Astrid Hendriks, Zhen Wang, Nina M. van Sorge, Lucy Pill-Pepe, Leslie Bautista, Ellen Chu, Jeroen D. C. Codée, Jeff Fairman, Neeraj Kapoor, Satoshi Uchiyama, Victor Nizet

**Affiliations:** 1Division of Host-Microbe Systems and Therapeutics, Department of Pediatrics, University of California San Diego, La Jolla, CA 92093, USA; sineadmccabe44@gmail.com (S.M.); ebjanes@health.ucsd.edu (E.B.); satoshi.uchiyama@cellothera.com (S.U.); 2Department of Medical Microbiology and Infection Prevention, Amsterdam UMC, University of Amsterdam, 1105 AZ Amsterdam, The Netherlands; a.hendriks2@amsterdamumc.nl (A.H.); n.m.vansorge@amsterdamumc.nl (N.M.v.S.); 3Leiden Institute of Chemistry, Leiden University, 2333 CC Leiden, The Netherlands; z.wang@lic.leidenuniv.nl (Z.W.); jcodee@chem.leidenuniv.nl (J.D.C.C.); 4Netherlands Reference Laboratory for Bacterial Meningitis, Amsterdam UMC, 1105 AZ Amsterdam, The Netherlands; 5Vaxcyte, Inc., San Carlos, CA 94070, USA; lucy.pepe@vaxcyte.com (L.P.-P.); leslie.bautista@vaxcyte.com (L.B.); elchu@ucdavis.edu (E.C.); jeff.fairman@vaxcyte.com (J.F.); neeraj.kapoor@vaxcyte.com (N.K.); 6Skaggs School of Pharmacy and Pharmaceutical Sciences, University of California San Diego, La Jolla, CA 92093, USA

**Keywords:** group A *Streptococcus*, *Streptococcus pyogenes*, group B *Streptococcus*, *Streptococcus agalactiae*, vaccine, C5a peptidase, virulence factor, cross-protective immunity, conjugate vaccine

## Abstract

Group B *Streptococcus* (*Streptococcus agalactiae* or GBS) is the leading infectious cause of neonatal mortality, causing roughly 150,000 infant deaths and stillbirths annually across the globe. Approximately 20% of pregnant women are asymptomatically colonized by GBS, which is a major risk factor for severe fetal and neonatal infections as well as preterm birth, low birth weight, and neurodevelopmental abnormalities. Current clinical interventions for GBS infection are limited to antibiotics, and no vaccine is available. We previously described VAX-A1 as a highly effective conjugate vaccine against group A *Streptococcus* that is formulated with three antigens, SpyAD, streptolysin O, and C5a peptidase (ScpA). ScpA is a surface-expressed, well-characterized GAS virulence factor that shares nearly identical sequences with the lesser studied GBS homolog ScpB. Here, we show that GBS C5a peptidase ScpB cleaves human complement factor C5a and contributes to disease severity in the murine models of pneumonia and sepsis. Furthermore, antibodies elicited by GAS C5a peptidase bind to GBS in an ScpB-dependent manner, and VAX-A1 immunization protects mice against lethal GBS heterologous challenge. These findings support the contribution of ScpB to GBS virulence and underscore the importance of choosing vaccine antigens; a universal GAS vaccine such as VAX-A1 whose formulation includes GAS C5a peptidase may have additional benefits through some measure of cross-protection against GBS infections.

## 1. Introduction

While a polyvalent capsular polysaccharide vaccine to protect against the serious *Streptococcus pneumoniae* disease was first licensed in 1977 and pneumococcal conjugate vaccines with efficacy in infants have been available since 2000 [1], there are currently no approved vaccines for two other encapsulated streptococcal pathogens that cause major infectious disease morbidity and mortality worldwide: group A *Streptococcus* (GAS, *S. pyogenes*) and group B *Streptococcus* (GBS, *S. agalactiae*). GAS is responsible for >600 million cases of “strep throat” or pharyngitis each year and can also produce serious invasive disease manifestations including sepsis, necrotizing fasciitis, and toxic shock syndrome, for which a resurgence has been documented in many countries across the world [2,3]. GAS is also the immunological trigger for acute rheumatic fever (ARF) and rheumatic heart disease (RHD), a predominant cause of cardiovascular morbidity and mortality throughout many low- and middle-income countries of the world [4]. GBS, which colonizes the gastrointestinal and urogenital tract of healthy adults including ~20% of pregnant women [5], remains a preeminent cause of neonatal sepsis and meningitis, resulting in ~150,000 stillbirths and infant deaths worldwide each year [6,7]. Additionally, GBS infections have been increasingly associated with a serious disease in vulnerable adults including the elderly, diabetics, pregnant women, and the immunocompromised. Bringing a safe and effective GAS or GBS vaccine to market would provide an immediate and enormous global public health impact. 

Challenges to GAS and GBS vaccine development are unique for each species, and reflect distinct scientific, logistic, safety, and health economic considerations. In the case of GAS, these challenges are as follows: (i) its surface capsule of high molecular weight hyaluronan is indistinguishable from a common human tissue constituent, (ii) its immunodominant surface-anchored M protein is antigenically hypervariable yielding more than 200 serotypes, and (iii) the post-infectious immune-mediated GAS sequelae of ARF/RHD raise safety concerns for indiscriminate antigen selection. Conversely, GBS expresses a limited set of unique capsular polysaccharides that upon conjugation to a carrier protein elicit robust opsonic antibody responses. Currently, advanced clinical trials are underway in pregnant women to ascertain if post-immunization transplacental IgG transfer can significantly protect infants against GBS invasive disease. The extent to which a maternal GBS conjugate vaccine can match the acceptance or efficacy of seasonal influenza vaccines or diphtheria/pertussis boosters in pregnancy is yet unknown. 

GAS vaccine development has gained recent momentum due to private and public initiatives and funding vehicles, including the WHO Product Development for Vaccines Advisory Committee (PD-VAC), the Strep A Vaccine Global Consortium (SAVAC), and the Combating Antibiotic-Resistant Bacteria Biopharmaceutical Accelerator (CARB-X), with several candidates now in advanced preclinical or early clinical development. Among these is VAX-A1 (Vaxcyte, Inc.), which is a multi-subunit conjugate vaccine centered around the universally conserved, highly abundant, and species-defining GAS cell-wall carbohydrate (GAC), engineered to lack its N-acetylglucosamine (GlcNAc) side chain, which has been linked to the autoimmune reactivity in ARF/RHD patients [8,9,10,11]. The remaining high-molecular weight GAC polyrhamnose backbone (GAC^PR^) is conjugated to the conserved GAS surface protein antigen SpyAD through a technology involving *cell-free* transcription/translation for non-natural amino acid incorporation and click-chemistry methods that retains full immunogenicity of this pathogen-specific carrier protein [12,13]. This bivalent conjugate vaccine is supplemented with two additional conserved GAS virulence factors to complete the VAX-A1 formulation: the secreted pore-forming toxin streptolysin O (SLO) that lyses host cells through membrane pore formation and the surface-anchored peptidase ScpA, which cleaves and inactivates the human complement-derived chemotaxin C5a, impeding neutrophil recruitment in vitro and in vivo [14,15,16] (Figure 1).

While GAS proteins SpyAD and SLO have no homologs in GBS, the neonatal pathogen expresses a peptidase, ScpB, with 97–98% sequence homology to GAS ScpA [17]. Recombinant ScpB cleaves and inactivates C5a in vitro, but its role in GBS virulence has not been evaluated in animal challenge models. We hypothesized that the high homology between GAS and GBS C5a peptidases may allow for the cross-reactivity of ScpA-elicited neutralizing antibodies, thereby neutralizing ScpB function. We further considered that VAX-A1 immunization may confer cross-protection against GBS challenge in addition to its primary embodiment as a GAS vaccine. 

GBS is a significant cause of sepsis in newborn infants, particularly in cases of early-onset infection resulting from fetal or newborn aspiration of contaminated amniotic fluids in utero or exposure to vaginal fluids during delivery. Here, we show using the targeted mutagenesis and heterologous expression of *scpB* that the encoded peptidase is both necessary and sufficient for the cleavage of human C5a. Moreover, ScpB contributes to GBS virulence in murine lung and systemic infection models. IgG antibodies elicited by the GAS homolog ScpA or conjugate vaccine candidate VAX-A1 display cross-reactivity to GBS ScpB, potentiating the opsonophagocytic killing of GBS by human neutrophils in a ScpB-dependent manner. VAX-A1 immunization conferred cross-protection to mice against the lethal sepsis challenge of two heterologous GBS strains. Overall, our data indicate a potential extended benefit of the VAX-A1 vaccine formulation toward another globally important invasive bacterial pathogen.

## 2. Materials and Methods

### 2.1. Bacterial Strains, Growth Conditions, and Genomic DNA Isolation

Group B *Streptococcus* (GBS) A909 serotype Ia (American Type Culture Collection ATCC: BAA-1138) was cultured in Todd-Hewitt broth (THB; Hardy Diagnostics) at 37 °C under static conditions until it reached the stationary phase. In some cases, THB was supplemented with antibiotics. The stationary phase cultures were then diluted in fresh THB and incubated at 37 °C until they reached the mid-logarithmic phase. GBS COH1 serotype III (ATCC: BAA-1176) was grown using similar growth conditions. To isolate the genomic DNA (gDNA) from GBS, the overnight culture of GBS was collected through centrifugation and resuspended in RLT-Plus Lysis Buffer (Qiagen). The resuspended cells were added to Lysis Matrix B (Qbiogene) and mechanically lysed using the Mini-Beadbeater-96 (Biospec Products). The bacterial gDNA was subsequently purified from the lysate using the NucleoSpin gDNA Clean-up Kit (Macherey-Nagel), following the manufacturer’s protocol.

### 2.2. Construction of the GBS scpB Insertional Mutant

Targeted insertional mutagenesis was performed using vector pHY304, following a previously described method [18]. A ~500 bp region of the GBS A909 *scpB* gene (Genbank: BA46302.1), containing the active site, was PCR-amplified using primers designed with overhangs for the restriction enzymes Xho1 and Xba1. The forward primer sequence was 5’-TG**CTCGAG**TATCAGAGATGCTATCAACTTGGGAGCTAAGGTG-3’, and the reverse primer sequence was 5’-GA**TCTAGA**GGAAGCCCTTGTCCTGATTGTCATAGATCA-3’. The amplified DNA product was then ligated into pHY304 and transformed into electrocompetent GBS A909 wild-type (WT) strain. Transformants were selected by erythromycin resistance (Em^R^) in agar at 30 °C. Single crossover insertions were induced by shifting the selected colonies to 37 °C with Em^R^ selection. The interruption of the *scpB* active site by the integrated pHY304 plasmid was confirmed through PCR analysis, the loss of IgG binding from immune serum (see details below), and the loss of ScpB enzymatic activity (see details below). The resulting *scpB* mutant (Δ*scpB*) strain was further cultured in media without antibiotic selection at 37 °C, and the colonies that lost Em^R^ were verified to regain ScpB expression (ScpB.Rev).

### 2.3. Heterologous Expression of GBS scpB in Lactococcus lactis

The complete *scpB* gene with EcoR1 and BamH1 overhangs was amplified with PCR from the A909 gDNA using primers scpB.For 5’-CT**GAATTC**TCATCATGAAAGGACGACACATTGC-3’ and scpB.Rev 5’-CA**GGATCC**CTATTTTTTAGTTTCTTTTTGGCGTTTTGTTTTAAATATA-3’. The PCR amplicon was then ligated into the *E. coli*-streptococcal shuttle expression vector pDCErm [19] and used to transform *L. lactis* MG1363 Electrocompetent Cells (Intact Genomics Inc., St. Louis, MO, USA).

### 2.4. C5a Cleavage Assay

C5a peptidase activity was assessed by incubating bacterial strains with recombinant human C5a (R&D Systems, 2037-C5-025/CF). Stationary phase cultures of GBS and *L. lactis* were resuspended at 2 × 10^9^ CFUs/mL and then incubated with 100 μg/mL of recombinant human C5a for 16 h at 37 °C. Following the incubation, NuPAGE LDS Sample Buffer 4× (Fisher, NP0007) was added to the sample supernatant and boiled for 5 min. Subsequently, the samples were run on a 4–12% Bis-Tris gel (Fisher) at 100 V and stained with SimplyBlue Safe Stain (Fisher, LC6065).

### 2.5. IgG Binding Assay

Pre-immune and immune serum samples were collected from rabbits that were immunized with recombinant GAS C5a peptidase and VAX-A1 by Vaxcyte, Inc. The rabbits received 5 μg of antigen per dose with a total of 3 doses administered 2 weeks apart. To detect Scp expression, bacterial strains were incubated with either pre-immune serum or immune serum for 1 h at room temperature after blocking with 10% heat-inactivated donkey serum. Following the serum incubation, Alexa Fluor 488-conjugated donkey anti-rabbit IgG antibody (IgG-AF488, Thermo-Fisher, Waltham, MA, USA) was added, and the samples were further incubated for 30 min in the dark. The level of IgG binding to bacteria was determined using flow cytometry analysis on a FACs Canto (BD) system. Geometric mean values of fluorescence intensity were analyzed using FlowJo Version 10.

### 2.6. Primary Human Neutrophil Opsonophagocytic Killing (OPK) Assay

The OPK assay was performed by following our previously described method [12]. Briefly, neutrophils were isolated from blood obtained from healthy human donors with their consent, as approved by the UC San Diego Human Research Protection Program IRB (Protocol #131002X), using Polymorphprep (Axis-Shield, Oslo, Norway) according to the manufacturer’s protocol. Neutrophils were adjusted to a concentration of 5 × 10^6^ cells/mL in RPMI 1640 tissue culture media, and the bacteria were pre-incubated with baby rabbit complement (PelFreez, #31061) and heat-inactivated fetal bovine serum (FBS, Cat# 97068-085, VWR International). GBS strains were grown to the mid-logarithmic growth phase of OD_600_ = ~0.4 and washed twice with sterile phosphate-buffered saline (PBS). Neutrophils were then added to the washed bacteria at a multiplicity of infection (MOI) of 0.1 bacteria per neutrophil in a 96-well flat-bottom tissue culture plate with the immune or control non-immune sera. The final concentration of components in the reaction mixture were 20% experimental or control serum (murine or rabbit), 2% fetal bovine serum, and 2% baby rabbit complement, with the remaining volume comprising bacteria and neutrophils in PBS. After incubation for the indicated times at 37 °C without shaking, neutrophils were lysed with ddH_2_0, and samples serially diluted in PBS and plated onto THB agar plates for colony-forming unit (CFU) enumeration. Sera from pre-immune rabbits were pooled and used as a control to measure non-specific, baseline bacterial killing by neutrophils for the rabbit antisera samples. Each serum or serum combination was tested in three biological replicates to ensure statistical confidence.

### 2.7. Methods for Glycan Analysis

The chemical synthesis of different oligomers mimicking the group B carbohydrate (GBC) and GAC polyrhamnose backbone has been described previously [20,21]. Magnetic streptavidin beads (Dynabeads M280 Streptavidin, ThermoFisher Scientific) were coated with biotinylated oligomers (0.17 mM) for 15 min at RT, followed by five washes with PBS using a magnetic sample rack. Beads were incubated with the three-fold serial dilutions of immunized rabbit sera (0.03–3%) for 20 min at 4 °C in PBS containing 0.1% bovine serum albumin (BSA, Serva) and 0.05% Tween-20. After washing, beads were incubated with AlexaFluor 647-conjugated goat anti-rabbit IgG antibodies (Southern Biotech), and IgG binding was analyzed using flow cytometry (BD FACS Canto). The geometric mean fluorescence intensity (GeoMFI) values were corrected for background binding (GeoMFI values for non-coated beads) and interpolated using a binding curve of anti-GBC polyclonal antibodies (0.01–10 µg/mL, ab53584, Abcam) to GBC-511 beads. Interpolated values were corrected for the dilution factor, and values from at least two dilutions were used to calculate the mean normalized IgG binding.

### 2.8. Murine Infection and Vaccination Models

Mouse experiments were conducted in accordance with ethical guidelines and approved by the UC San Diego Institutional Animal Care and Use Committee (Protocol #S00227M) following standard veterinary and animal care practices. The mice were housed in a specific pathogen-free facility under a 12-hour light/dark cycle and kept in pre-bedded corn cob disposable cages (Innovive), provided with a 2020X diet (Envigo), and had access to acidified water. Wild-type female CD-1 mice (Charles River, Wilmington, MA, USA) were used for virulence (8–10 weeks of age) and vaccine (beginning at 5 weeks of age) studies. In the pneumonia model, we used a method adapted from our studies with *Staphylococcus aureus* and *Pseudomonas aeruginosa* [22]. Briefly, animals were anesthetized with 100 mg/kg of ketamine and 10 mg/kg xylazine, then infected with 3 × 10^8^ CFUs of WT or ∆*scpB* GBS by direct intratracheal inoculation. Mice were monitored on a heating pad until fully recovered from anesthesia. At twenty hours post-infection, mice were euthanized with CO_2_ and their lungs were excised, homogenized, serially diluted in sterile PBS, and plated on THA for enumeration. For the sepsis model, mice were briefly anesthetized with isoflurane, then infected retro-orbitally with 1 × 10^8^ CFU of WT GBS, ∆*scpB* GBS or ScpB.Rev in sterile PBS. The mice were monitored daily to assess morbidity and mortality. Kaplan–Meier survival curves were used to analyze the data, and statistical significance was determined using the log-rank Mantel–Cox test. The immunization process involved administering doses of VAX-A1 every 14 d, starting at 5 weeks of age. Each intramuscular immunization consisted of 5 μg of each vaccine antigen along with 50 μL of Alhydrogel 2% aluminum hydroxide adjuvant (Invivogen), prepared per manufacturer’s instructions. Two weeks after the final immunization, mice were infected with 1 × 10^8^ CFU of GBS A909 or COH1 through intraperitoneal (i.p.) injection. The mice were monitored daily to assess morbidity and mortality. Kaplan–Meier survival curves were again used to analyze the data, and statistical significance was determined using the log-rank Mantel–Cox test.

### 2.9. Statistical Analyses

Statistical Analysis was conducted with GraphPad Prism. Two groups were compared using two-tailed Students *t* test. Three or more groups were compared with one-way ANOVA with multiple comparisons. Mortality curves were assessed with log-rank Mantel–Cox tests. Values *p* < 0.05 were deemed significant. Flow cytometry analysis was performed with FlowJo. Gates were drawn using unstained and single stained controls.

## 3. Results

### 3.1. GBS ScpB Is a Functional C5a Peptidase

To assess the virulence function of GBS C5a peptidase, we created a mutant strain (∆*scpB*) from the wild-type (WT) serotype Ia GBS strain A909 using plasmid insertional (Campbell-type) mutagenesis (Figure 2A, see methods for details). To generate the WT revertant strain ScpB.Rev, we subjected the ∆*scpB* mutant to serial passaging and identified colonies that lost their antibiotic resistance (Erm^R^) marker. Furthermore, we cloned the GBS *scpB* gene into a plasmid vector to express it heterologously in the non-pathogenic bacterium *L. lactis*, resulting in strain LL[pScpB] for gain-of-function analysis (Figure 2A). To validate the presence or absence of ScpB enzymatic activity, we incubated the panel of strains with recombinant human C5a: WT GBS and LL[pScpB] exhibited the cleavage of the chemotactic peptide, whereas the GBS ∆*scpB* mutant and untransformed *L. lactis* did not (Figure 2A).

### 3.2. GBS ScpB Contributes to Animal Virulence

To investigate the role of ScpB in GBS virulence, we employed an intratracheal murine challenge model of infection. Our findings revealed that mice infected with the GBS ∆*scpB* mutant had notably fewer bacteria in their lungs 20 h after infection, in comparison to those infected with WT GBS or Scp.Rev GBS (Figure 2B). This result indicates that *scpB* plays a critical role in in vivo pathogenesis. Additionally, we conducted an intravenous sepsis model of infection, where mice infected with the GBS ∆*scpB* mutant demonstrated a significantly improved survival compared to those infected with WT GBS or ScpB.Rev, further substantiating ScpB as a virulence factor (Figure 2C). 

### 3.3. IgG Produced against GAS ScpA Cross-Reacts with ScpB on the GBS Surface

Given the significant homology between GAS ScpA and GBS ScpB, we sought to explore whether the surface-expressed GBS ScpB could be recognized by IgG antibodies generated against the GAS homolog ScpA. To achieve this, we immunized New Zealand white rabbits with recombinant GAS C5a peptidase (ScpA), and post-immune serum was collected one week after the final immunization (Figure 3A). Pre-immune serum obtained before vaccination served as the negative control. In our surface binding experiments using WT GBS, we observed a notably higher binding of the anti-ScpA immune serum compared to the pre-immune serum. Importantly, binding was dependent on the expression of ScpB, as the GBS ∆*scpB* mutant displayed a minimal increase in anti-ScpA IgG binding (Figure 3A). WT *L. lactis* did not show IgG binding to either pre-immune or immune serum (Figure 3B). However, LL[pScpB] demonstrated an enhanced IgG binding when exposed to the anti-ScpA immune serum relative to the pre-immune serum (Figure 3B,C). These collective results indicate that GBS ScpB can bind and be recognized by antibodies produced against GAS ScpA, underscoring the cross-reactivity of ScpB in the context of GBS and GAS infections.

### 3.4. IgG Produced against GAC Polyrhamose Cross-Reacts Weakly with Oligosaccharide Sequences in the Group B Cell Wall Carbohydrate (GBC)

Unlike the linear polyrhamnose backbones of GAC (Figure 4A) and the group carbohydrate of group C *Streptococcus* (GCS) [23], the species-defining and chromosomally encoded [24] GBS cell wall carbohydrate (GBC) possesses a complex multi-antennary structure (Figure 4B) [25]. However, a prominent feature of the phospho-octasaccharide subunits of GBC antennary structure are tri-rhamnose terminal motifs [23,26]. Thus, we explored whether antisera produced against GAC^PR^ cross-reacted with the rhamnose moieties in these GBC motifs. Three terminal rhamnose-containing oligosaccharide structures from GBC, namely, 510 (an octasaccharide repeat), 505 (the pentapeptide cap), and 511 (comprising 510 + 505 joined together) (Figure 4B), were synthesized, biotinylated, and coupled to streptavidin-coated latex beads. These beads were then probed with anti-GAC^PR^ immune serum (Figure 4C), with non-coated beads serving as a negative control and GAC^PR^-coated beads serving as a positive control. Normalized to a binding curve established with anti-GBS polyclonal IgG antibodies (Figure 4D), we found that IgG antibodies in anti-GAC^PR^ antisera recognized structure 510 in a concentration-dependent manner but to a modest extent, and structures 505 and 511 to a significantly lesser degree (Figure 4E,F). We conclude that weak but appreciable cross-reactivity of anti-GAC^PR^ IgG antibodies to rhamnose-containing structures in GBC, the conserved GBS cell wall carbohydrate, exists.

### 3.5. VAX-A1 Antisera Recognition of GBS Occurs Principally through Conserved Streptococcal C5a Peptidase Antigens

The VAX-A1 GAS conjugate vaccine formulation comprises SpyAD-GAC^PR^, SLO, and C5a peptidase (ScpA) with an alum adjuvant [12] (Figure 1). Next, we investigated if antibodies were elicited in response to the VAX-A1 multicomponent GAS vaccine candidate (**F**igure 1) cross-protected against live GBS and whether the high degree of ScpA/ScpB homology was sufficient to explain such cross-protection, as neither SLO nor SpyAD share homology with proteins encoded in the GBS genome. We found that VAX-A1 immunization generates antibodies that recognize GBS bacteria in a ScpB-dependent manner, as binding seen in WT and ScpB.Rev strains is absent in the ∆*scpB* mutant (Figure 5A). We found that WT GBS and the ScpB.Rev strains had significantly increased IgG binding after incubation with the VAX-A1 serum relative to the pre-immune rabbit serum, while ∆*scpB* GBS did not (Figure 5A,C). Furthermore, *L. lactis +* pScpB significantly bound IgG in the presence of the VAX-A1 antisera relative to the pre-immune serum, which was not seen in wildtype *L. lactis* (Figure 5B,C). These findings indicate that antibodies produced by the GAS conjugate vaccine VAX-A1 can bind and recognize GBS ScpB.

Since VAX-A1 is a conjugate vaccine, we examined whether the cross-reactivity of GBS could be due to the recognition of the other protein components. We generated rabbit immune serum produced against GAS SpyAD alone and SpyAD conjugated to the polyrhamnose backbone (SpyAD-GAC^PR^). Upon incubation with either immune serum, we found that increased IgG binding by WT GBS occurred only in the presence of ScpA immune serum and that binding could be abrogated with the loss of ScpB expression (Figure 5D). Together these results show that the cross-reactivity of GBS to VAX-A1 immune sera is specific for and dependent on C5a peptidase (ScpB) expression.

Opsonophagocytosis by neutrophils is a major defense mechanism against bacterial infections; thus, we examined the efficacy of VAX-A1 immune serum in potentiating neutrophil killing of GBS. Freshly purified human neutrophils were incubated with heat-inactivated pre-immune, VAX-A1, or ScpA immune serum in the presence of WT or ∆*scpB* GBS (Figure 6A). VAX-A1 and ScpA immune sera significantly increased the bactericidal activity of human neutrophils against WT GBS relative to pre-immune serum. The survival of the ∆*scpB* GBS mutant was equivalent in the presence of VAX-A1 and ScpA immune sera compared to pre-immune serum, indicating that enhanced killing by neutrophils was dependent on the presence of ScpB (Figure 6A).

Bolstered by significant homology in cross-reactivity and function between ScpB and ScpA, we tested if the vaccination with VAX-A1 could protect mice against the lethal challenge of clinically important GBS strains. For this, five-week-old CD-1 mice were immunized intramuscularly with three doses of VAX-A1 administered two weeks apart (Figure 6B). Fourteen days after the final immunization, mice were challenged intraperitoneally with an ~LD_50_ dose of two well-characterized invasive clinical isolates, representing the two most common invasive serotypes in neonatal infection, i.e., WT strains GBS A909 (serotype Ia) and COH1 (serotype III), and their survival was monitored for 8 days. As shown in Figure 6C, immunization with VAX-A1 provided a significantly higher protection when challenged with a lethal dose of both GBS strains. These results suggest that in addition to the protective efficacy against lethal GAS challenge [12], vaccination with VAX-A1 may provide additional protection against another streptococcal pathogen of tremendous clinical importance, namely, GBS.

## 4. Discussion

Group A and B streptococci exhibit a high level of sequence homology in their C5a peptidase genes, namely, *scpA* and *scpB*, respectively [17]. C5a peptidase and its unique target specificity were first identified in GBS [17,27,28], where an ability to bind fibronectin was later reported and postulated to contribute to mucosal colonization [28]. However, the roles of C5a peptidase in immune evasion and systemic infection have been more extensively studied in GAS [14,29,30,31]. In the present study, we further investigated the role of ScpB in GBS virulence and explored the biological significance of its homology with ScpA. Through loss- and gain-of-function experiments, we found ScpB was not only necessary for the GBS degradation of human C5a, but also sufficient to endow a heterologous nonpathogenic strain of *L. lactis* with the ability to cleave the chemokine. We also demonstrate for the first time experimentally that ScpB contributes to in vivo virulence in two murine models of infection. These results collectively highlight the importance of ScpB in GBS-mediated pathogenicity and suggest its potential as a target for intervention strategies against GBS-associated infections.

The remarkable 98% sequence similarity between GAS *scpA* and GBS *scpB* produced the strong possibility of shared binding site recognition for antibodies between these two antigens. We found that live GBS could indeed be bound by IgG antibodies produced against GAS ScpA, and this interaction was found to be ScpB-dependent. This interaction held a biological significance for vaccine design, as antisera against GAS ScpA enhanced the opsonophagocytic killing of GBS by human neutrophils, but only in the presence of ScpB. Neutralizing antibodies against C5a should have the potential dual benefit of enhancing opsonophagocytosis and blocking C5a cleavage, thereby allowing a greater influx of neutrophil to the sites of infection. Intriguingly, although approximately 20% of human clinical strains of GBS lack detectable C5a peptidase enzymatic activity as a result of *scpB* sequence polymorphisms [32], an *scpB* gene is nevertheless present in nearly all isolates (~98%), suggesting additional key functions beyond chemokine inactivation that warrant its evolutionary conservation [33].

A major long-term public health benefit of a safe and effective GAS vaccine will be a reduction in the global incidence of RHD, a leading cause of cardiovascular morbidity and mortality in many regions of the developing world [34,35]. The Global Burden of Disease (GBD) Study calculated that 33.4 million people were afflicted with RHD globally in 2015, with 319,400 annual deaths resulting from the critical post-infectious immune-mediated complication of GAS infection [36]. RHD is approximately twice as common in women compared to men [37], and the presence of mitral stenosis in pregnancy is especially ominous [38]. Early in pregnancy, cardiovascular and hematological changes result in hyperdynamic circulation, leading to a heightened risk of heart failure that steadily increases up to 24 weeks before plateauing until 30 weeks [39]. In one report, among 50 pregnant patients hospitalized with heart disease in Senegal, 46 had RHD, and the maternal mortality rate was 34% [40]. In a study of 281 cases of maternal heart disease from India, the most prevalent condition in pregnancy was RHD, observed in 195 cases (69.4%), with isolated mitral stenosis being the predominant subtype in 75 cases (26.7%) [41]. RHD in pregnancy is associated with additional adverse neonatal outcomes including preterm birth and intrauterine growth retardation, reinforcing that women of childbearing age are an important target group for immunization [42]. 

Children are the priority target for GAS vaccination, as they experience the highest incidence of streptococcal pharyngitis, impetigo, and autoimmune complications. Persistence of vaccine-induced immunity in girls entering childbearing age could extend a public health benefit to a critical demographic for passive protection against neonatal GBS infection. A recent analysis of the global health burden of maternal and newborn GBS disease estimated more than 200,000 infants developed early-onset infection (0–6 days) and approximately 160,000 experienced late-onset infections (6–89 days) in 2020, resulting in estimated 91,000 annual deaths, with the highest deaths occurring in sub-Saharan Africa and Asia [6]. Moreover, approximately 46,000 stillbirths and more than 500,000 excess premature births were attributed to maternal GBS colonization [6]. Thus, the effective cross-protection of a vaccine between GAS and GBS could markedly impact maternal and infant health.

## 5. Conclusions

The quest for a universal GAS vaccine has been challenging, and, thus far, no candidates have received clinical approval and advanced to the market [43]. In previous research, we reported a significant protection in preclinical studies produced by VAX-A1, a multivalent protein conjugate GAS vaccine containing SpyAD, streptolysin O, and ScpA [12]. We observed IgG cross-binding and enhanced opsonophagocytic killing mediated by the anti-ScpA antibody cross-recognition of GBS ScpB, leading to the finding that VAX-A1 immunization could potentially confer cross-protection against the neonatal pathogen. Indeed, in a murine sepsis model, mice vaccinated with three doses of VAX-A1 were shielded from infections caused by two GBS strains. Thus, human GAS vaccines that incorporate C5a peptidase as an antigen might offer unexpected benefits in terms of cross-protection against GBS infections. Although several GAS vaccine efforts have focused on targeting the dominant surface epitope, M protein, a subset of recent conjugate vaccines in preclinical development have included C5a peptidase as a candidate antigen in their multivalent formulations [12,44,45,46]. If these particular GAS vaccines continue to progress successfully, it may be fruitful to explore any secondary impact on the incidence of newborn GBS infections during Phase III clinical trials or through post-marketing analysis.

## Figures and Tables

**Figure 1 vaccines-11-01811-f001:**
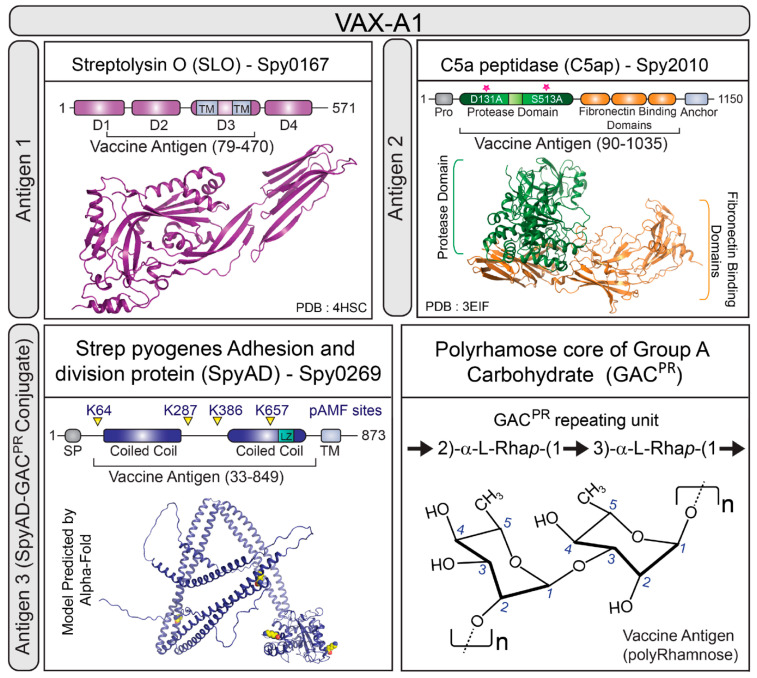
Components of group A *Streptococcus* vaccine candidate VAX-A1. The VAX-A1 vaccine candidate is composed of three GAS antigens. Antigen 1: Streptolysin O (SLO), with indicated transmembrane (TM) regions. Antigen 2: C5a peptidase (C5ap), with red asterisks indicating D131A and S513A mutations. Antigen 3: the *S. pyogenes* adhesion and division protein (SpyAD), with the position of p-azidomethyl phenylalanine (pAMF) replacements shown, conjugated to the polyrhamnose core of the group A carbohydrate (GAC) (SpyAD-Gac^PR^).

**Figure 2 vaccines-11-01811-f002:**
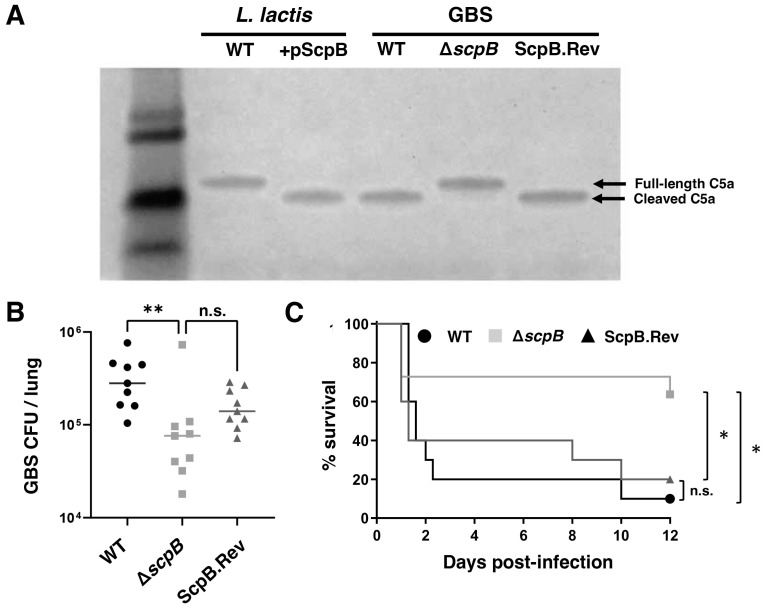
ScpB is necessary and sufficient for human C5a cleavage and promotes in vivo virulence in murine models of infection. (**A**) The GBS Δs*cpB* mutant was generated by insertional mutagenesis, and *Lactococcus lactis* was transformed with the plasmid (+pScpB) for gain-of-function analysis. Western blot showing human C5a cleavage by WT GBS is absent in the Δs*cpB* mutant and restored in the revertant strain ScpB.Rev. Expression of pScpB confers C5a cleavage phenotype to *L. lactis*. (**B**) Lung colony forming units (CFUs) from mice infected intratracheally with 3 × 10^8^ CFUs of WT GBS, ∆*scpB* mutant, or ScpB.Rev in sterile PBS. CFUs were assessed 20 h post-infection. Each dot represents one mouse, n = 8 per group. (**C**) Mortality curves in mice retro-orbitally infected with 1 × 10^8^ CFUs of WT GBS, ∆*scpB* mutant, or ScpB.Rev in sterile PBS; n = 10 per group. n.s., not significant, * *p* < 0.05, ** *p* < 0.01.

**Figure 3 vaccines-11-01811-f003:**
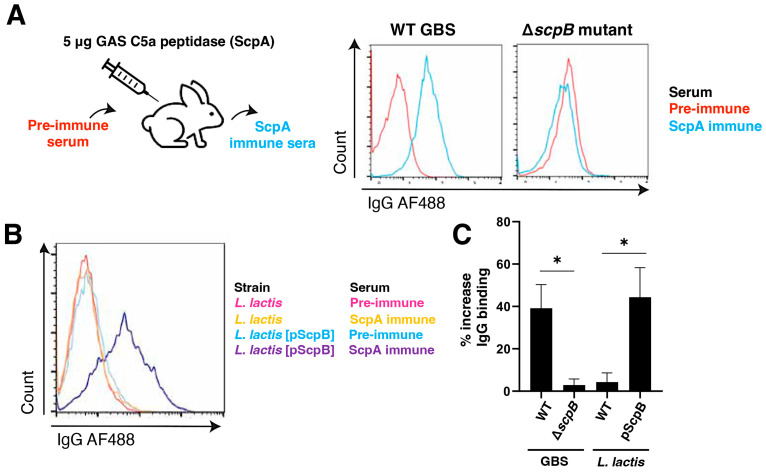
IgG antibodies elicited by GAS ScpA cross-react with GBS ScpB. (**A**) New Zealand white rabbits were immunized with 5 ug of recombinant GAS C5a peptidase, ScpA. Pre-immune and ScpA immune sera were provided by Vaxcyte, Inc. Representative histograms showing IgG-AF488 binding to WT GBS or ∆*scpB* GBS in the presence of pre-immune or ScpA immune serum. (**B**) Representative histograms depicting IgG-AF488 binding to WT *L. lactis* or *L. Lactis* + pScpB in the presence of pre-immune or ScpA immune serum. (**C**) Quantification of **A** and **B**. * *p* < 0.05.

**Figure 4 vaccines-11-01811-f004:**
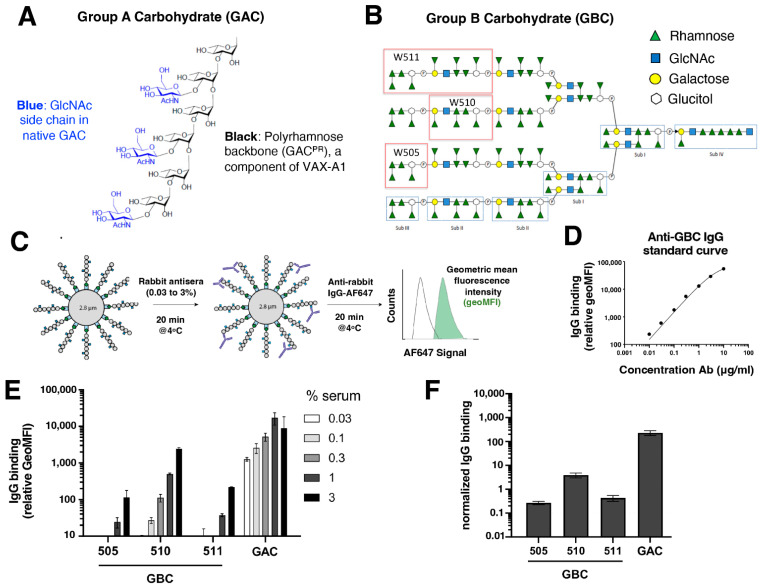
Cross-reactive potential of GAC polyrhamnose-specific IgG antibodies for group B carbohydrate. (**A**) Composition of repeating structure of GAC. (**B**) Composition of group B carbohydrate (GBC), with synthetic oligomers (505, 510, 511) indicated. (**C**) Experimental setup to analyze IgG binding to GBC beads in GAC^PR^-immunized rabbit sera. (**D**) Concentration-dependent binding of anti-GBC polyclonal rabbit IgG antibodies (0.01–10 ug/mL) to beads coated with the 511 oligomers. (**E**) IgG binding to synthetic GBC/GAC-coated beads in different concentrations of immunized rabbit sera, and geoMFI values have been corrected for background binding (relative GeoMFI). (**F**) Normalized IgG binding to GBC/GAC beads, and dotted line represents the lower limit of quantification.

**Figure 5 vaccines-11-01811-f005:**
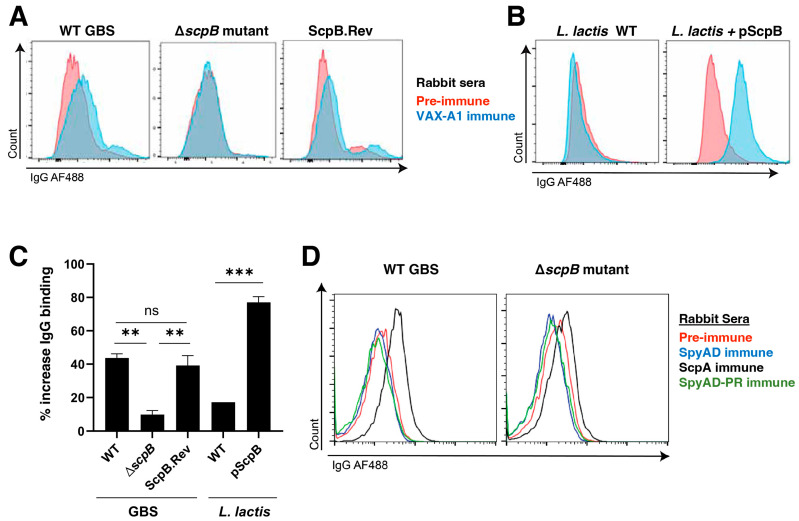
ScpB is cross-reactive to ScpA-specific IgG antibodies elicited by VAX-A1. New Zealand white rabbits were immunized with (**A**–**C**) 5 ug of VAX-A1 or (**D**) its components: C5a peptidase, SpyAD alone, or SpyAD conjugated to the polyrhamnose backbone (SpyAD^RH^). Pre-immune and immune sera were provided by Vaxcyte, Inc. (**A**) Representative histograms showing IgG-AF488 binding to WT GBS, the ∆*scpB* mutant, or ScpB.Rev in the presence of the pre-immune or VAX-A1 immune serum. (**B**) Representative histograms showing IgG-AF488 binding to WT *L. lactis* or *L. lactis* transformed with pScpB in the presence of the pre-immune or VAX-A1 immune serum. (**C**) Quantification of **A** and **B**. (**D**) Representative histograms showing IgG-AF488 binding to WT GBS and the isogenic ∆*scpB* mutant in the presence of pre-immune, ScpA immune, SpyAD immune, or SpyAD^RH^ immune serum. n.s. not significant, ** *p* < 0.01, *** *p* < 0.001.

**Figure 6 vaccines-11-01811-f006:**
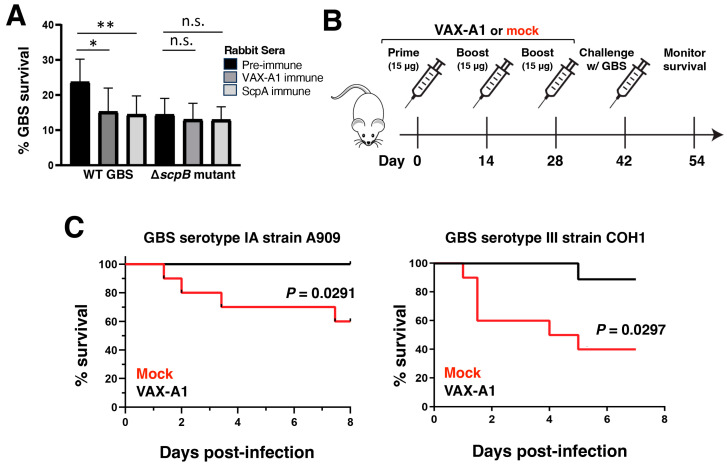
ScpB cross-reactivity mediates increased opsonophagocytic killing and in vivo protection against two GBS strains. (**A**) CFUs showing survival of WT GBS and ∆*scpB* GBS opsonized by pre-immune, ScpA, and VAX-A1 immune sera in the presence of healthy human neutrophils. (**B**) Mice were vaccinated with PBS (mock) or 15 ug of VAX-A1 (5 ug per antigen) intramuscularly on Days 0, 14, and 28. (**C**) On Day 42, mice were infected with 1 × 10^8^ CFU GBS A909 (serotype IA) or COH1 (serotype III) by intraperitoneal (i.p.) injection and monitored daily for morbidity and mortality, n = 10 per group. n.s., not significant. * *p* < 0.05, ** *p* < 0.01.

## Data Availability

Data generated or analyzed during this study are included in this published article.

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
