# Peer review of "The Group A Streptococcal Vaccine Candidate VAX-A1 Protects against Group B *Streptococcus* Infection via Cross-Reactive IgG Targeting Virulence Factor C5a Peptidase"

_vaccines, 2023, doi:10.3390/vaccines11121811_

Round 1
Reviewer 1 Report
Comments and Suggestions for Authors
Review of the manuscript: The Group A Streptococcal Vaccine Candidate VAX-A1 Protects 2 Against Group B Streptococcus Infection via Cross-Reactive 3 IgG Targeting Virulence Factor C5a Peptidase
The overall quality of the manuscript is very high. I have some observations that can be found below.
Line 130. Correct reference (it should be placed before the final point).
Lines 165-166. I suggest including the "(IgG AF 488)” after the description of such an antibody.
Line 176. Correct “an” bacteria…
Line 177. Put a comma before “and heat-inactivated…”
Line 185. The word “were” should be placed between “neutrophils” and “lysed”.
Line 214. Correct reference (remove italics and include a final point after the reference number).
Results
General comment: statistical validation is missing in almost all presented data. I suggest authors present a proper statistical analysis of results concerning figures: 2 B and C; 3; 4 C – F; 5; 6A. Interestingly, figures 6 B and C show statistical validation.
Fig 2A. What does exactly mean this image of protein electrophoresis? L. lactis +pScbB as well as GBS WT, and GBS ScpB.Rev proteins have smaller MW than the others. Are these human C5a cleavage products? And the others, human C5a? What are the MWs of them? I suggest to make the presentation of such results clearer.
Fig 2B. What do these two asterisks mean?
Fig 2C. What is the X axis? Days? Hours? It must be stated.
Fig 3B. Histogram colors are difficult to understand, so I suggest to modify them.
Line 333. Is it correct: “C5ap-specific”?
Lines 348-349. Revise the sentence. Punctuation or a word is missing.
Fig 6A. What do these asterisks mean?
Fig 6B. This figure is not a result, therefore, it should not be presented as part of the results. In addition, the image does not correspond to the written text: “For this, five-week old CD-1 mice were immunized intramuscularly with three doses of VAX-A1 separated two weeks apart (Figure 6B). Fourteen days post final immunization, mice were challenged intraperitoneally…”
Line 401. I think authors would like to write “vaccine” instead of “disease”.
Lines 413-415. Authors should discuss better the question about the GAS vaccine for women of childbearing age: although both pathogens can impact pregnancy outcomes, it is crucial to state that the GAS vaccine priority target is childhood, the most vulnerable for GAS infections and immunological-based diseases.
Author Response
REVIEWER #1
The overall quality of the manuscript is very high. I have some observations that can be found below.
Thank you for the kind words. We have incorporated your valuable feedback into the revised manuscript as described below in a point-by-point manner.
Line 130. Correct reference (it should be placed before the final point).
The position of the reference citation [14] was moved and placed correctly in front of the period.
Lines 165-166. I suggest including the "(IgG AF 488)” after the description of such an antibody.
The shorthand designation IgG-AF488 was added after the antibody description.
Line 176. Correct “an” bacteria…
The typo was corrected to “and the bacteria were ...”
Line 177. Put a comma before “and heat-inactivated…”
This correction is no longer needed with the edit above. The new sentence reads:
Neutrophils were adjusted to a concentration of 5 x 106 cells/mL in RPMI 1640 tissue culture media, and the bacteria were pre-incubated with baby rabbit complement (PelFreez, #31061) and heat-inactivated fetal bovine serum (FBS, Cat# 97068-085, VWR International).
Line 185. The word “were” should be placed between “neutrophils” and “lysed”.
This correction was made, and it now reads “neutrophils were lysed”
Line 214. Correct reference (remove italics and include a final point after the reference number).
The reference was changed to plain text and a period added after it.
Results
General comment: statistical validation is missing in almost all presented data. I suggest authors present a proper statistical analysis of results concerning figures: 2 B and C; 3; 4 C – F; 5; 6A. Interestingly, figures 6 B and C show statistical validation.
We have updated the methods section to include a description of the statistical analysis utilized for each experiment and added statistical markings to figures and figure legends.
Fig 2A. What does exactly mean this image of protein electrophoresis? L. lactis +pScbB as well as GBS WT, and GBS ScpB.Rev proteins have smaller MW than the others. Are these human C5a cleavage products? And the others, human C5a? What are the MWs of them? I suggest to make the presentation of such results clearer.
This figure is significant because it shows the loss of C5a cleavage activity in the mutant. Human C5a uncleaved is 12-14 kDa which is seen in the image and is consistent with previous reports (Monk PN, Scola AM, Madala P, Fairlie DP. Function, structure, and therapeutic potential of complement C5a receptors. Br J Pharmacol. 2007;152:429-48) cleaved C5a in the WT, L. lactis + scpB and scpB.Rev. The uncleaved C5a at a higher MW which is only seen in the WT L. lactis and ΔScpB mutant. This is now indicated with arrows more clearly in the revised figure.
Fig 2B. What do these two asterisks mean?
We have updated the figure legend to include the statistical information that **p<0.01.
Fig 2C. What is the X axis? Days? Hours? It must be stated.
We have updated the figure and the legend to specify that the X axis is Days post-infection.
Fig 3B. Histogram colors are difficult to understand, so I suggest to modify them.
We have modified the colors to be more easily discriminated.
Line 333. Is it correct: “C5ap-specific”?
The sentence was revised and now reads “ScpB is cross-reactive to ScpA-specific IgG antibodies elicited by VAX-A1”
Lines 348-349. Revise the sentence. Punctuation or a word is missing.
The sentence was revised and now reads as follows: “Opsonophagocytosis by neutrophils is a major defense mechanism against bacterial infections, thus we examined the efficacy of VAX-A1 immune serum in potentiating neutrophil killing of GBS”
Fig 6A. What do these asterisks mean?
We’ve updated the figure legend to include the statistical information that **p<0.01.
Fig 6B. This figure is not a result, therefore, it should not be presented as part of the results. In addition, the image does not correspond to the written text: “For this, five-week old CD-1 mice were immunized intramuscularly with three doses of VAX-A1 separated two weeks apart (Figure 6B). Fourteen days post final immunization, mice were challenged intraperitoneally…”
Figure 6B provides a visual description of the vaccination and infection schedule for Figure 6C. Mice were given three immunizations every two weeks, on days 0, 14, and 28. Fourteen days after the final vaccination, mice were infected with GBS (day 42) and monitored for survival.
Line 401. I think authors would like to write “vaccine” instead of “disease”.
Yes, thank you for pointing out this obvious typo, which has now been corrected.
Lines 413-415. Authors should discuss better the question about the GAS vaccine for women of childbearing age: although both pathogens can impact pregnancy outcomes, it is crucial to state that the GAS vaccine priority target is childhood, the most vulnerable for GAS infections and immunological-based diseases.
Agree. The start of the paragraph has been rephrased as follows: "Children are the priority target for GAS vaccination, as they experience the highest incidence of streptococcal pharyngitis, impetigo, and autoimmune complications. Persistence of vaccine-induced immunity in girls entering into childbearing age could extend a public health benefit into the critical demographic for passive protection versus neonatal GBS infection."
Reviewer 2 Report
Comments and Suggestions for Authors
The manuscript describes a study showing that GBS C5a peptidase (ScpB) cleaves human complement factor C5a and contributes to disease severity in murine models of pneumonia and sepsis. Furthermore, antibodies elicited by GAS C5a peptidase (ScpA), which is a well-characterized GAS virulence factor and shares nearly identical sequences with ScpB , bind to GBS in an ScpB-dependent manner and that immunization with VAX-A1 vaccine which contains ScpA protects mice against lethal GBS heterologous challenge. The manuscript is well written, studies well executed and interesting data is presented. The manuscript represents an interesting contribution in the field
Author Response
We thank the Reviewer for their generous comments and appreciation of our work.
Reviewer 3 Report
Comments and Suggestions for Authors
The authors provide an intriguing and well-supported argument that a vaccine against GAS that includes C5a peptidase could provide cross-protection against group B streptococcus as well. My major suggestion to the authors is that future work in this area would be strengthened by inclusion of Serotype II GBS as well as Serotypes Ia and III in future experiments. Adding this third serotype would cover 60-70% of serotypes from invasive infant GBS disease from the ABCs surveillance system in the U.S. Serotype II has been gradually expanding as a cause of invasive GBS in the U.S. was the most prevalent serotype overall in ABCs data in 2021 and the most common serotype among isolates from invasive infant in 2017, 2019 and 2021.
Other more minor comments include:
1) Line 51, suggest "significant global resurgence" may be overly strong language - consider deleting "signficant global" particularly since you state "many countries" later in sentences
2) Line 54, recommend "Low and middle-income countries" rather than "developing regions".
3) Lines 82-83, please provide a reference for the statement "which has been linked to the autoimmune reactivity in ARF/RHD patients."
4) Figure 1, needs definition of other acronyms used (e.g. pAMF, TM)
5) Line 176, seems to be a word missing or misspelled with "an bacteria"
6) Lines 200 and 202, geoMFI and GBC need to be defined. Please check that all abbreviations are defined first time used
7) Line 237, ScpB,Rev should be defined earlier
8) Lines 254-256, this sentence belongs in background not results section
9) Lines 264-266, suggest this sentence goes in discussion/conclusions section rather than results
10) Figure 3, unclear why results for SCpB.Rev are not included in Figure 3a if included in quantification in Figure 3c
11)Line 292, suggest using acronym SDSE rather than GCS since SDSE also incorporates group G as well as group C
12) Line 305, should this be 511 not 510 in this sentence?
13) Figure 4f, unclear why there is less IgG binding for 511 if it comprises 510. Is this because there is relatively less 510 oligomer?
14) Figure 5, the apparent different scales on the y-axis makes comparisons challenging. Is it possible to make these graphs larger and have the units on the y-axis more visible?
15) Figure 5, the abbreviation in the first sentence of the legend appears incorrect (C5ap)
16) Lines 353-356. The intended meaning of this sentence, in particular "survived equivalently" is unclear. Suggest rewriting for clarity
17) Lines 397-399. This sentence is a little confusingly worded in trying to understand why 20% of strains account for 2% of isolates. In part this might be due to the word "clinical" in front of strains, which makes it sound like strain is being used interchangeably with isolates. Suggest rewording for clarity
18) Line 408-409, unclear why only this one study from Senegal is cited to support this statement as other studies with larger sample sizes are available for citation
19) Line 427, there appears to be a redundant "of" in front of IgG in this sentence
Author Response
REVIEWER #3
The authors provide an intriguing and well-supported argument that a vaccine against GAS that includes C5a peptidase could provide cross-protection against group B streptococcus as well. My major suggestion to the authors is that future work in this area would be strengthened by inclusion of Serotype II GBS as well as Serotypes Ia and III in future experiments. Adding this third serotype would cover 60-70% of serotypes from invasive infant GBS disease from the ABCs surveillance system in the U.S. Serotype II has been gradually expanding as a cause of invasive GBS in the U.S. was the most prevalent serotype overall in ABCs data in 2021 and the most common serotype among isolates from invasive infant in 2017, 2019 and 2021.
Other more minor comments include:
1) Line 51, suggest "significant global resurgence" may be overly strong language - consider deleting "signficant global" particularly since you state "many countries" later in sentences.
Agreed. We have deleted the words “significant” and “global” as recommended.
2) Line 54, recommend "Low and middle-income countries" rather than "developing regions".
We have made this recommended modification.
3) Lines 82-83, please provide a reference for the statement "which has been linked to the autoimmune reactivity in ARF/RHD patients."
We have cited the following 4 references and adjusted the numbering of all the in-text citations and reference list accordingly.
- Goldstein, I., Rebeyrotte, P., Parlebas, J., and Halpern, B. (1968). Isolation from heart valves of glycopeptides which share immunological properties with Streptococcus haemolyticus group A polysaccharides. Nature 219, 866–868.
- Shulman, S.T., Ayoub, E.M., Victorica, B.E., Gessner, I.H., Tamer, D.F., and Hernandez, F.A. (1974). Differences in antibody response to streptococcal antigens in children with rheumatic and non-rheumatic mitral valve disease. Circulation 50, 1244–1251.
- Appleton, R.S., Victorica, B.E., Tamer, D., and Ayoub, E.M. (1985). Specificity of persistence of antibody to the streptococcal group A carbohydrate in rheumatic valvular heart disease. J. Lab. Clin. Med. 105, 114–119.
- Galvin, J.E., Hemric, M.E., Ward, K., and Cunningham, M.W. (2000). Cytotoxic mAb from rheumatic carditis recognizes heart valves and laminin. J. Clin. Invest. 106, 217–224.
4) Figure 1, needs definition of other acronyms used (e.g. pAMF, TM).
The definitions of transmembrane (TM) and p-azidomethyl phenylalanine (pAMF) are provided.
5) Line 176, seems to be a word missing or misspelled with "an bacteria"
The typo was corrected to “and the bacteria were”
6) Lines 200 and 202, geoMFI and GBC need to be defined. Please check that all abbreviations are defined first time used.
The definitions of geometric mean fluorescence intensity (GeoMFI) and Group B Carbohydrate (GBC)—at the start of the paragraph—are provided.
7) Line 237, ScpB,Rev should be defined earlier
ScpB.Rev was described earlier in the methods section, so we have removed the parentheses which made it seem here to be a new definition. Now the sentence simple states: “To generate the WT revertant strain ScpB.Rev …”
8) Lines 254-256, this sentence belongs in background not results section
This sentence: “GBS is a significant cause of sepsis in newborn infants, particularly in cases of early-onset infection resulting from fetal or newborn aspiration of contaminated amniotic fluids in utero or exposure to vaginal fluids during delivery.” was moved to the beginning of the last sentence of the introduction,
9) Lines 264-266, suggest this sentence goes in discussion/conclusions section rather than results.
The sentence: “These results collectively highlight the importance of ScpB in GBS-mediated pathogenicity and suggest its potential as a target for intervention strategies against GBS-associated infections.” was removed from the results section and moved to the end of the first paragraph in the discussion.
10) Figure 3, unclear why results for SCpB.Rev are not included in Figure 3a if included in quantification in Figure 3c
We have updated Figure 3c to reflect the results presented in Figure 3a (not included SCpBrev).
11)Line 292, suggest using acronym SDSE rather than GCS since SDSE also incorporates group G as well as group C.
It is true that SDSE includes GGS and GCS, however, only GCS shares the polyrhamnose backbone of the cell wall antigen with GAS. Thus, we have eliminated reference to S. dysgalactiae subsp. equisimilis for clarity, just denoting Group C Streptococcus (GCS).
12) Line 305, should this be 511 not 510 in this sentence?
Yes, this has been corrected.
13) Figure 4f, unclear why there is less IgG binding for 511 if it comprises 510. Is this because there is relatively less 510 oligomer?
It is unclear, but that was the result. It is possible that the low binding 505 component sterically inhibits access to the 510 domain in 511.
14) Figure 5, the apparent different scales on the y-axis makes comparisons challenging. Is it possible to make these graphs larger and have the units on the y-axis more visible?
We’ve updated the graphs to be easier to read.
15) Figure 5, the abbreviation in the first sentence of the legend appears incorrect (C5ap)
Yes, this has been corrected to ScpA.
16) Lines 353-356. The intended meaning of this sentence, in particular "survived equivalently" is unclear. Suggest rewriting for clarity.
This was reworded as follows: “Survival of the ∆scpB GBS mutant was equivalent in the presence of VAX-A1 and ScpA immune serum compared to pre-immune serum …”
17) Lines 397-399. This sentence is a little confusingly worded in trying to understand why 20% of strains account for 2% of isolates. In part this might be due to the word "clinical" in front of strains, which makes it sound like strain is being used interchangeably with isolates. Suggest rewording for clarity.
The sentence was reworded as follows: “Intriguingly, although approximately 20% of human clinical strains of GBS lack detectable C5a peptidase enzymatic activity as a result of scpB sequence polymorphisms [28], an scpB gene is nevertheless present in nearly all isolates (~98%), suggesting additional key functions beyond chemokine inactivation that warrant its evolutionary conservation ”
18) Line 408-409, unclear why only this one study from Senegal is cited to support this statement as other studies with larger sample sizes are available for citation
We have revised this section and added two references as below:
Early in pregnancy, cardiovascular and hematological changes result in hyperdynamic circulation, leading to a heightened risk of heart failure that steadily increases up to 24 weeks before plateauing until 30 weeks [39]. In one report, among 50 pregnant patients hospitalized with heart disease in Senegal, 46 had RHD and the maternal mortality rate was 34% [40]. In a study of 281 cases of maternal heart disease from India, the most prevalent condition in pregnancy was RHD, observed in 195 cases (69.4%), with isolated mitral stenosis being the predominant subtype in 75 cases (26.7%) [41].
[39] Ruys, T.P.; Roos-Hesselink, J.W.; Hall, R.; Subirana-Domènech, M.T.; Grando-Ting, J.; Estensen, M.; Crepaz, R.; Fesslova, V.; Gurvitz, M.; De Backer, J.; Johnson, M.R.; Pieper, P.G. Heart failure in pregnant women with cardiac disease: data from the ROPAC. Heart. 2014, 100, 231-8.
[41] Konar, H.; Chaudhuri S. Pregnancy complicated by maternal heart disease: a review of 281 women. J. Obstet. Gynaecol. India. 2012, 62, 301-6.
19) Line 427, there appears to be a redundant "of" in front of IgG in this sentence
Yes, the redundant “of” was removed.